# Satellite-Derived Bathymetry in Support of Maritime Archaeological Research—VENμS Imagery of Caesarea Maritima, Israel, as a Case Study

**Gerardo Diaz** [1,*], **Yoav Lehahn** [2] and **Emmanuel Nantet** [1]

1 Department of Maritime Civilizations, School of Archaeology and Maritime Cultures, University of Haifa, Haifa 3103301, Israel; enantet@univ.haifa.ac.il
2 Department of Marine Geosciences, Leon H. Charney School of Marine Sciences, University of Haifa, Haifa 3103301, Israel; ylehahn@univ.haifa.ac.il
* Correspondence: gerardo@bgu.ac.il

**Abstract:** Deriving bathymetry by means of multispectral satellite imagery proves to be a replicable method, offering high-resolution coverage over large areas while keeping costs low. Maritime archaeologists often require bathymetric mapping at a high resolution and with a large spatial coverage. In this paper, we demonstrate the implementation of SDB in maritime archaeology using high-resolution (5 m/pixel) data from Vegetation and Environment monitoring on a New Micro-Satellite (VENμS) imagery. We focus on the area of the Roman harbour of Sebastos, located at Caesarea Maritima along the Israeli coast of the Eastern Mediterranean. For extracting SDB, we take an empirical approach, which is based on the integration of satellite imagery and sonar depth measurements, resulting in a blue-green band ratio algorithm that provides reliable results up to a water depth of 17 m. Comparison with in situ depth measurements yielded an RMSE of 0.688 m. The SDB mapping is complemented by satellite-based identification of above- and below-water rocks. The presented approach can readily be replicated in other regions using various types of multispectral satellite imagery, particularly when only coarse bathymetric sonar data are available, thus substantially contributing to our ability to perform maritime archaeological research.

**Keywords:** satellite-derived bathymetry; VENμS; rock identification; maritime archaeology

## 1. Introduction

Bathymetric mapping of spatial variations in water depth provides the most basic information on the morphology of the seafloor [1]. Its significance and applications are found in multiple fields, including navigational planning [2], offshore industry assessments [3,4], coral reef mapping [5,6], the study of coastal processes [7], creating hydrodynamic models [8], and marine life research [9]. In the realm of maritime archaeology, bathymetry serves as a fundamental tool for understanding shipwreck scours, site formation processes [10,11], and palaeolandscape reconstructions [12–14], as well as for conserving, protecting, and preserving underwater cultural heritage [15,16]. Other applications include improving planning visits to archaeological sites by tourist divers and optimising for locating mechanical devices (e.g., hoses, dredgers, weights, water jets, etc.), focusing on either protecting or excavating the archaeological artifacts left on the site [15].

Traditionally, bathymetry is derived from ship-based measurements. The most common methods rely on acoustic systems, capable of retrieving depths ranging from very shallow waters to depths exceeding 190 meters such as single-beam [17,18] and multibeam echo-sounders [19,20]. Aerial options can also be found in the form of Light Detection and Ranging (LiDAR), which uses a green light (in bathymetric projects) in the form of a pulsed laser to measure ranges (variable distances) and generates 2D and 3D models of

the seafloor geomorphology [21] or by integrating echosounders into UAV drones [22]. Although nowadays these methods are easy to handle, quick to deploy, compact, and capable of collecting very-high- to ultra-high-resolution bathymetric data at speeds of 5–7 knots covering areas up to 5 sq. km per day, they still require the presence of an operator on site, and remain restricted to local authorisations and limited by political boundaries.

Satellite-derived bathymetry stands as a cost-effective complementary instrument, which can be employed using empirical methods [23–26], look-up tables [27–29], or semi-analytical models [29–32]. Lately, stereo approaches [33,34], machine learning [35–37], and deep learning techniques [38] have been implemented as well. The utilisation of multi-spectral satellite imagery for the extraction of bathymetry maps relies on the exponential wavelength-dependent nature of visible and near-infrared light attenuation along the water column [39–41], resulting in SDB maps up to a 30 m depth in clear water conditions [42].

Due to its inherent advantages, SDB has been increasingly used in various fields such as safety of navigation charting [43], coastal zone management and development [38], and coastal mapping [44], as well as submarine pipelines and cable laying [45]. Maritime archaeologists require high-resolution (in terms of time and space) bathymetric mapping and unbounded spatial coverage for planning underwater prospections, documenting and recording submerged features, performing preservation maneuvers, and conducting excavation campaigns. A few papers can be found in this realm [13,46,47]; similarly, the European project ITACA uses satellite data to derive bathymetry for the heritage management of coastal archaeological sites [48]. In this paper, we aim to demonstrate and consequently promote the implementation of SDB in maritime archaeological research, focusing on high-resolution (5 m/pixel) data from Vegetation and Environment monitoring on a New Micro-Satellite (VENμS) imagery, which is available free of charge to the scientific community.

Launched in August 2017, VENμS is a recent initiative between the Israeli Space Agency (ISA) and the French Space Agency (CNES) aimed at developing, manufacturing, and operating a new Earth-observing satellite using a superspectral camera that provides high-resolution imagery in terms of both space and time [49]. The general missions and objectives of VENμS are mainly focused on scientific purposes, from land use and vegetation mapping to watercolour characterisation for applications in continental hydrology and coastal oceanography [50]. Since it was launched, VENμS has planned five missions, each of which is scientific (S), technological (T), or a combination of both. Technological missions do not capture any images. The first mission (VM01) started in November 2017 and ended in October 2020; the second (VM02) started in November 2020 and finished in August 2021. During this mission, the satellite orbit was lowered to 410 km. VM03 occurred in September 2021 and lasted for one month. Mission 4 was aimed at changing VENμS's orbit at 560 km, starting in August and finishing in October 2021. The present mission VM05's sensing period started in March 2022, providing imagery since mid-December 2022 at a four-meter resolution [51], and it will continue until July 2024.

The extraction of SDB from VENμS imagery involves the implementation of an empirical ratio model [23] of multispectral bands, which is then compared with the available ship-based sonar bathymetry measurements conducted off the Israeli coast of the Eastern Mediterranean Sea.

## 2. Study Area

The study area selected for this research encompasses the Roman harbour of Sebastos (Figure 1a, Area 'A'), located at Caesarea Maritima and north of the modern port of Hadera, extending along the Israeli coast in the Eastern Mediterranean. The former was built in the year 21 BCE by King Herod and was probably functioning by 16 BCE [52]. When completed, it flourished due to the maritime trade between Rome and Alexandria [53]. King Herod's port city project included an intimate palace built from carved local calcareous-cemented aeolianite rocks (aka kurkar) [53]. The harbour of Sebastos consists of three basins [54,55]: the western or outer harbour was built in the open sea using a mixture of lime, aggregate,

and volcanic ash (aka pozzolana) as hydraulic concrete, which was cast into wooden work forms, and these were sunk into their intended positions [54,56]; the middle or intermediate basin, constructed on a coastal kurkar ridge, harnessed its natural geomorphology; and the inner basin, currently inland, was founded on a shallow marine feature on the lee side of a partly submerged coastal kurkar ridge.

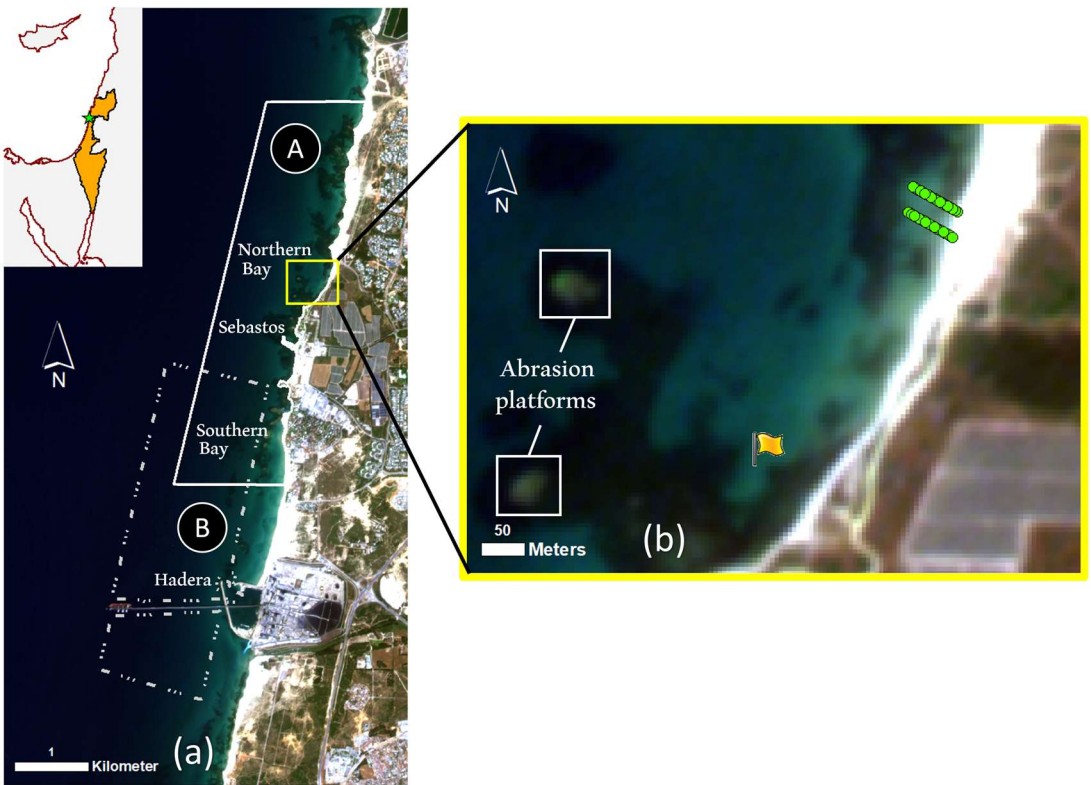

**Figure 1.** A true-colour VENμS satellite image of the study area: (**a**) macro- and micro-localisation of Caesarea Maritima and the port of Hadera. Solid line delineates the area where SDB is mapped (area 'A'). Dashed line delineates the areas used for the extraction of the empirical SDB algorithm through comparison with sonar data (area 'B', Figure 2). The insert shows the location of the study area in the Eastern Mediterranean. (**b**) A zoom-in on the area used for validation. The yellow flag and green dots mark, respectively, the locations of Caesarea Shipwreck's hull and two survey lines, where ground-truth data were collected. White boxes mark the locations of the abrasion platforms.

Bays closed with abrasion platforms (Figure 1a,b) are also present within the region [55,57,58]. The abrasion platforms are located ca. 950 m north of the Herodian harbour (Figure 1a,b). These are the products of marine corrosion that creates surf notches on exposed rocky coasts under long, stable sea-level conditions [59]. The southern bay, located ca. 1000 m south of the port of Sebastos (Figure 1a), has been considered the most natural protected feature of the harbour system of Caesarea. Its exploitation was an inherent necessity due to the unfavorable topography of the Levantine coast, which lacks the benefit of sheltered coves and protruding peninsulas [57].

The Hadera modern port is located around 3.7 km south of the ancient harbour of Sebastos (Figure 1a, area 'B'). This port was completed in the early 1960s, and it currently specialises in coal and fuel oil products and serves the two adjacent power stations [60,61]. This harbour was selected due to the availability of the ground-truth and sonar data necessary to develop the equation required to generate SDB maps. Analogously, the harbour of Sebastos was chosen because it is a well-studied port [52–58,62], due to the presence of in situ depth measurements for data validation (Figure 1b), and because of the absence of high-resolution bathymetry in the vicinity.

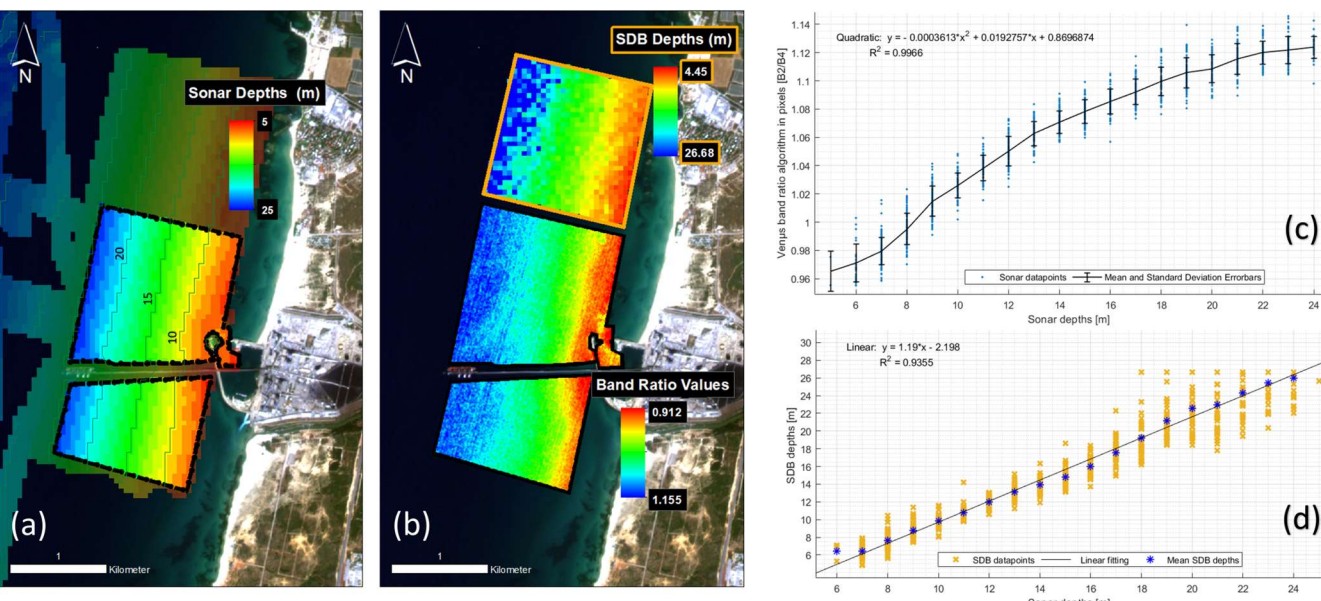

**Figure 2.** Relationship between water depth and satellite-derived bathymetry. (**a**) Sonar depth values (50 m/pixel) and (**b**) band ratio values (5 m/pixel) in the vicinity of the port of Hadera. (**c**) Band ratio (Equation (1)) and (**d**) SDB plotted against water depths measured using the sonar. In panels c and d, the satellite data are resampled to fit the spatial resolution of the sonar data (of 50 m/pixel).

## 3. Materials and Methods

The approach we took was based on the synergy between (1) the multispectral data from the VENµS satellite, (2) large-scale ship-based sonar depth measurements, (3) small-scale water depth measurements from archaeological expeditions, and (4) drone-based imaging. The satellite and sonar data were jointly used for the extraction of an SDB algorithm, which was then implemented to construct a high-resolution (5 m/pixel) SDB map of the study area. The map is complemented by satellite-based identification of the above-water and underwater rocks. The SDB and rock identification algorithms were validated against high-resolution data from archaeological expeditions and drone-based imaging, respectively.

### 3.1. VENµS Satellite Data

A satellite-derived bathymetry map was extracted using VENµS imagery downloaded from https://venus.bgu.ac.il/venus/ (accessed on 21 May 2020). The VENµS satellite flies in a near-polar, Sun-synchronous orbit; it is currently configured to revisit the same locations every two days (except under some circumstances, such as during polar nights), and it has the capability to observe any site under the same observation angle (to minimise directional effects) [49]. VENµS incorporates 12 narrow (≤40 nm) spectral bands [B1–B12] with a radiometric resolution of 10 bits (Table 1). It also provides three processing levels of products (L1, L2, and L3). L1 contains exclusively top-of-atmosphere reflectance (TOA) values; L2 and L3 provide surface reflectance, which means that they are already atmospherically corrected. L1 and L2 consist of images captured on a single date from a single angle acquisition, whereas L3 comprises 10-day composite and single-angle acquisitions. The images acquired during Mission 1 (2017–2020) have a spatial resolution of 5 m/pixel, while those collected during Mission 5 (2022–2024) have a spatial resolution of 4 m/pixel. Deriving a reliable SDB map requires the utilisation of satellite images that are taken under suitable environmental conditions, including a calm sea, minimal sun glint, no turbid waters, and sky free of clouds over the area of interest. Here, we used L2 VENµS imagery meeting these criteria acquired over the study area on 21 May 2020.

**Table 1.** Band specification summary for VENμS satellite imagery.

| Bands | λ Min (nm) | λ Max (nm) | λ Central (nm) | Bandwidth |
|-------|-----------|-----------|---------------|-----------|
| B1 | 383.9 | 463.9 | 423.9 | 40 |
| B2 | 406.9 | 486.9 | 446.9 | 40 |
| B3 | 451.9 | 531.9 | 491.9 | 40 |
| B4 | 515 | 595 | 555 | 40 |
| B5 | 579.7 | 659.7 | 619.7 | 40 |
| B6 | 589.5 | 649.5 | 619.7 | 40 |
| B7 | 636.2 | 696.2 | 666.2 | 40 |
| B8 | 678 | 726 | 702 | 30 |
| B9 | 725.1 | 757.1 | 741.1 | 24 |
| B10 | 766.2 | 798.2 | 782.2 | 16 |
| B11 | 821.1 | 901.1 | 861.1 | 40 |
| B12 | 888.7 | 928.7 | 908.7 | 20 |

*3.2. Sonar Data*

The sonar bathymetric data were obtained using a Kongsberg EM 1002 multibeam echosounder, placed on board the R/V Eziona, and the ELAC SeaBeam 3050 N multibeam system from Wärtsilä, deployed on board the R/V Mediterranean Explorer [63]. The sounding depth total accuracy for the EM 1002 multibeam is approximately 10 cm in shallow waters; the expected total system accuracy is 0.2% of the depth from vertical up to 45 degrees, 0.3% of the depth up to 60% degrees, and 0.5% of the depth between 60 and 70 degrees [64]. Seabeam 3050 N's accuracy is down to 2 cm [65].

The data were collected between 2001 and 2006 and further processed and analysed by the Geological Survey of Israel and Israel Oceanographic and Limnological Research, obtaining a bathymetric 3D model measuring 50 m × 50 m (grid size) at a 1 m (vertical) resolution. The survey covered a total area of 15,462 km$^2$ across the Israeli deep basin and its proximal seafloor, as well as the slope and the continental shelf. While being highly reliable, the relatively coarse horizontal and vertical resolution, coupled with their inability to cover near-shore areas and regions proximate to harbours, make it impractical for maritime archaeological applications. The sonar depth range for the Hadera port goes from 5 to 25 m (Figure 1a, area 'B' and Figure 2a).

It is important to note that while the sonar bathymetry data are used as a benchmark for regional bathymetry, they do not take into account the temporal changes associated with different processes such as sea level rise, sediment transport, and coastal erosion. Since the time difference between the sonar and satellite data collection is in the order of 20 years, given the fact that the sonar data are collected at water depths larger than 5 m, it is fair to assume that, during that period, bathymetry would not have exhibited large-scale changes that would have manifested in the relatively coarse vertical and spatial resolution sonar data.

*3.3. Ground-Truth Data*

Validation of the SDB retrieval was undertaken using data collected during two archaeological expeditions near the abrasion platforms (Figure 1b). One set of data includes 10 depth measurements of a hull's merchantman shipwreck excavated in December 2017 (located at 32.50802°N/34.8936°E) in Caesarea under the authorisation of the Israel Antiquities Authority and the Israel Nature and Parks Authority. These points were collected employing a shore-based Leica Ts06 Plus Total Station (an electronic surveying instrument that allows precise measurements of angles and distances) and three skilled divers, ensuring prism stability. This technique is further described in the NAS handbook [66] (p. 92).

The second dataset consists of 60 measurements arranged in two lines perpendicular to the shoreline collected on 5 July 2023. These measurements were conducted using a dumpy level (it consists of a telescope mounted on a stable base, which ensures horizontal

alignment and rotation for precise measurements of height differences) positioned on the shoreline. The measurements commenced at the shoreline, progressing into the sea at 2 m intervals. To ensure precision, a long open-reel measuring tape was straightened. A team consisting of two divers, two swimmers, and two individuals on the shore collaborated for each measurement. Standard hand signals were established to indicate points measured, points requiring revaluation, and the completion of a measurement line. This process was reiterated at a distance of 30 m. Obtaining these sets of data posed challenges, as it necessitated a proficient team of divers and favorable sea conditions. Consequently, in situ data collection might exhibit certain discrepancies. However, it gives the present investigation the opportunity to utilise a tool for checking the methodology in the field.

Validation of the satellite-derived rock identification was carried out using aerial images collected using a DJI Mavic Mini drone hovering at a 50 m altitude on 10 February 2021 over the abrasion platforms. The drone images were processed and analysed using the 3D modelling software Agisoft Metashape version 1.8.4 and converted into orthomosaics, from which above-water and underwater rocks were manually identified.

### 3.4. Empirical Model for SDB Retrieval

Retrieval of the SDB was undertaken based on Stumpf's algorithm [23] (p. 550), which relies on the ratio between the natural logarithm of the reflectance spectral bands corresponding to low (blue) and higher (green) absorption of light by the water:

$$z_{relative} = \frac{ln(nR_w(\lambda_i))}{ln(nR_w(\lambda_j))}, \tag{1}$$

where $\lambda_i$ and $\lambda_j$ correspond to the blue and green wavelengths, respectively, and $n$ is generally set to 1000 to ensure a positive outcome [67–69]. As can be seen in Figure 2, up to a bathymetric depth of 24 m, this band ratio is closely linked with the water depth, nicely capturing the spatial bathymetry patterns measured using sonar (Figure 2a,b).

Retrieval of the actual water depths from the band ratio values was undertaken through statistical analysis of the linkage with the sonar-measured water depths in the part of the study area that is covered by both datasets (Figure 2a). The values obtained from the band ratio first underwent low-filter convolution to smooth and decrease the disparity between the pixel values. Then, all the pixel values were resampled from a 5 to 50 m pixel size to match the sonar's spatial resolution using the nearest resampling technique because it is the fastest resampling approach, and it does not change the cells' values. The maximum spatial error is one-half of the cell size [70], which does not pose a significant difference for the following processing stages. Next, comparing the sonar bathymetric measurements with different band combinations revealed that the best agreement is achieved when using the ratio between band 2 (blue) and band 4 (green) (Figure 2c). Transforming the values of the band ratio into the estimated depths yields the following equation for SDB extraction from the specific VENµS data:

$$Z_{sdb} = -0.0003613x^2 + 0.0192757x + 0.8696874 \tag{2}$$

with $x$ being the band ratio described in Equation (1). Subsequently, this equation was validated by comparing the estimated satellite-derived depths and the corresponding sonar data collected over the northern section of the port of Hadera (delineated by a golden frame in Figure 2b). This comparison yielded a correlation coefficient ($R^2$) of 0.9355 (Figure 2d). A dot map is then generated by applying the equation across the entire area covered by the satellite imagery using the GIS mapping software ArcMap 10.8.2. The final SDB map (matching VENµS's spatial resolution) is obtained using the kriging interpolation method as suggested by Amante [71] and Conger [72] for bathymetric purposes.

Comparing the satellite and sonar–depth measurements in the port of Hadera (Figure 2) shows a distinct difference between water depths of 17 m and 24 m, with a

root mean square error (RMSE) of 0.949 m and 2.037 m, respectively (Table 2). Accordingly, we set the upper limit of our SDB retrieval to be 17 m.

**Table 2.** Comparison between sonar and SDB depths over the area used for validation of SDB equation in the port of Hadera (golden framed box in Figure 2b). The satellite data are resampled to fit the spatial resolution of the sonar data (of 50 m/pixel). RMSE values are calculated from the entire datasets.

| Sonar Depth (m) | Mean SDB Values (m) | Std Dev. SDB Values (m) | Diff. in Percentage (%) | Depth Difference (m) |
|---|---|---|---|---|
| 6 | 6.407 | 0.990 | 6.783 | 0.407 |
| 7 | 6.414 | 0.791 | 8.371 | −0.586 |
| 8 | 7.662 | 0.955 | 4.225 | −0.338 |
| 9 | 8.758 | 0.821 | 2.689 | −0.242 |
| 10 | 9.812 | 0.731 | 1.880 | −0.188 |
| 11 | 10.762 | 0.726 | 2.164 | −0.238 |
| 12 | 11.995 | 0.669 | 0.042 | −0.005 |
| 13 | 13.098 | 0.836 | 0.754 | 0.098 |
| 14 | 13.909 | 0.828 | 0.650 | −0.091 |
| 15 | 14.781 | 0.997 | 1.460 | −0.219 |
| 16 | 16.018 | 1.081 | 0.113 | 0.018 |
| 17 | 17.548 | 1.356 | 3.224 | 0.548 |
| 18 | 19.200 | 2.368 | 6.667 | 1.200 |
| 19 | 21.157 | 2.162 | 11.353 | 2.157 |
| 20 | 22.581 | 2.694 | 12.905 | 2.581 |
| 21 | 22.973 | 3.051 | 9.395 | 1.973 |
| 22 | 24.312 | 2.425 | 10.509 | 2.312 |
| 23 | 25.449 | 1.803 | 10.648 | 2.449 |
| 24 | 25.975 | 1.399 | 8.229 | 1.975 |
| | | | RMSE [6–17 m] = 0.949 m | |
| | | | RMSE [6–24 m] = 2.037 m | |

### 3.5. Satellite-Derived Rock Identification

In addition to SDB mapping, the VENμS data were used to identify rocks above water (RAW) and rocks below water (RBW). First, a land/sea mask was applied using a Normalised Difference Water Index (NDWI) [73]; then, identification of RAW and RBW was achieved by employing a combination of band differencing, Principal Component Analysis (PCA), and Maximum Likelihood Classification (MLC). Identifying the rocky features above the water (RAW) in the near-shore region was achieved by subtracting the red edge [band 8] from the red band [band 7]. For rocks below the water (RBW), the process was more challenging. Initially, all the raw bands were visually analysed, and those exhibiting both types of rocks were selected. For the VENμS imagery, bands 2, 3, 4, 7, and 8 were chosen. Note that alternative bands can also demonstrate efficacy, and it is not imperative to use all 12 VENμS bands at once. Next, a PCA technique was applied to reduce the data dimensionality. Following, training polygons were drawn, each corresponding to five different categories: submerged rocks, rocks above the water, deep water regions, shallow areas, and swash zones. Subsequently, only underwater rocks are isolated and converted into polygons. This thorough approach facilitated the identification and classification of rocky features above and below the water within the specified area of interest.

### 4. Results

The empirical SDB and rock identification algorithms were used to map a 6.2 km² area of interest overlapping with our archaeological site. The different single-band VENμS images used in the processes are displayed in Figure 3a, showing some emerging features in the near-shore environment. As can be seen in Figure 3b,c, subtracting band 8 from

band 7 highlights the RAW features yielded, with favorable results obtained when setting a threshold at three times the standard deviation of the mean value.

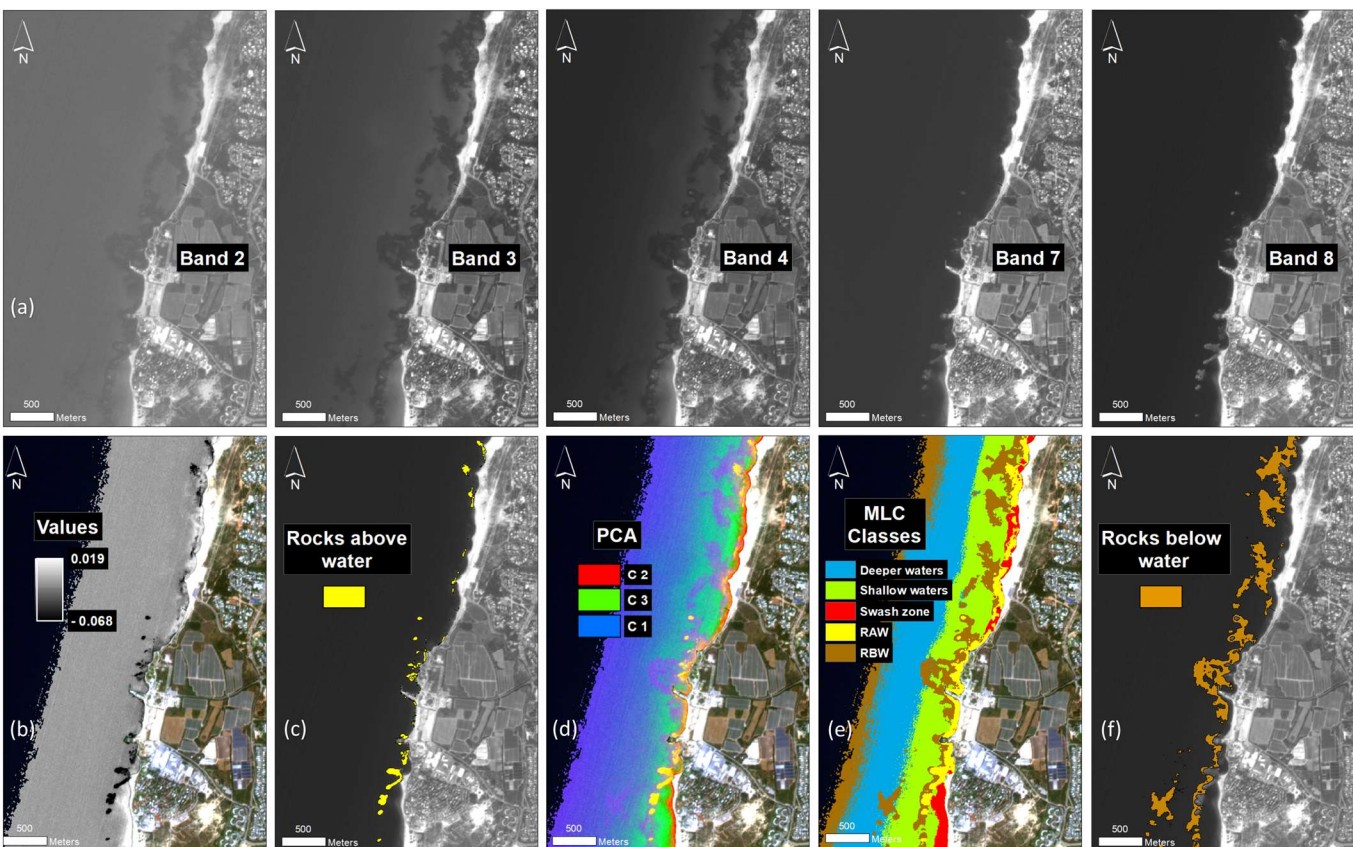

**Figure 3.** Rocky feature identification process: (**a**) VENμS satellite bands 2, 3, 4, 7, and 8 (from left to right); (**b**) results of subtracting band 8 from band 7. (**c**) Map of RAW. (**d**) PCA obtained from bands 2, 3, 4, 7, and 8. (**e**) MLC of five distinct classes. (**f**) Map of RBW.

The results from the PCA, MLC classification, and RBW reclassification are shown in Figure 3d–f. A constraint was found in deeper regions where pixel values from underwater rocks and deeper water areas exhibited nearly identical spectral responses, limiting the method's effectiveness to distances of less than 1 km from the shoreline. Nonetheless, it remains suitable for archaeological objectives within a near-shore region, particularly in shallow areas.

The ability to map the above- and below-water rocks from the satellite imagery is visually corroborated in Figure 4, where the subsets of the RAW and RBW maps are overlaid onto orthomosaics of these two types of rocks. The spatial comparison reveals a remarkable agreement between the areas identified as RAW and RBW in the satellite imagery and the orthomosaics. Naturally, satellite-based rock identification is limited in its ability to delineate, in detail, the boundaries of the rocks due to its 5 m resolution, which, compared to the drone imagery, is coarse. The manual detection of these features in a RGB image would be time-consuming and could lead to misinterpretations; instead, both approaches offer a semi-automatic and suitable rock detection procedure for rocks above and below the water level.

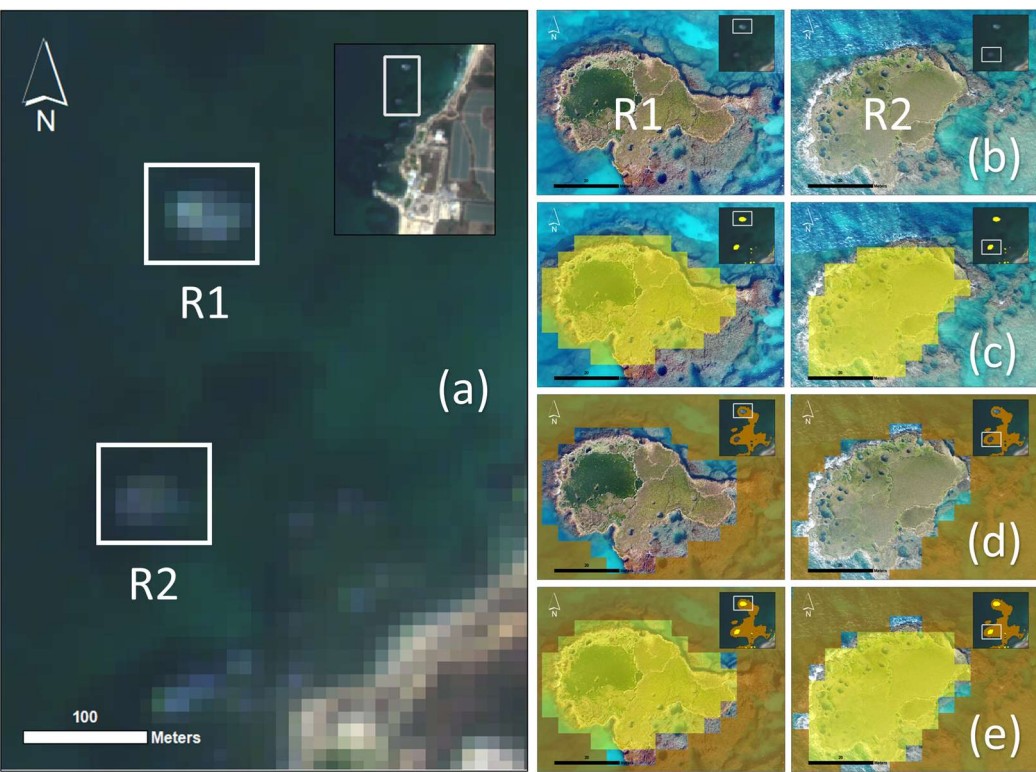

**Figure 4.** Rock identification within the harbour system of Caesarea. (**a**) True-colour satellite images of the abrasion platforms, labelled R1 and R2. (**b**) Orthomosaics of both of them. (**c**–**e**) The same as b but with areas identified as RAW and RBW in the satellite imagery coloured in yellow and brown, respectively.

Based on the methodology described above, a point grid matrix of nx4 containing the point identification (ID), latitude (X), longitude (Y), and depths (Z) derived from the satellite imagery has been used as a source for producing a high-resolution (5 m/pixel) bathymetric map covering 6.2 km$^2$ of the study area (Figure 5). Both rock layers (RAW and RBW) were overlaid on top to create a thorough depiction of the near-shore landscape of the harbour system of Caesarea Maritima. Figure 5 shows that shallow areas (0–2.5 m) are found along almost the entire shore, except in the harbour of Sebastos itself, where the depths vary from 2.5 to 7.5 m in the inner and outer basins, respectively. In yellow, RAW are displayed, showing the two abrasion platforms in the north and around four other abrasion platforms in the south.

To evaluate the SDB's performance at water depths lower than 3 m, the satellite depths were compared with the in situ depth measurements taken in the shipwreck's hull in 2017 and along two lines in 2023 (Figure 6, Tables 3 and 4). This comparison yielded an RMSE of 0.688 m, showing the ability of our approach to provide reliable SDB that can be used in support of maritime archaeological research. For comparison, Westley [13] (p. 8) reported an RMSE of 2.56 m using Sentinel 2 data (10 m/pixel) and polynomial fitting, and Li et al. [74] (p. 5) reported RMSE values between 1.22 m and 1.86 m using the Planet Dove satellites (4 m/pixel). We note that while the SDB values have been well validated for water depths below 3 m (Figure 6, Tables 3 and 4) and between 6 and 17 m (Figure 2d and Table 2), for the depth range of 3–6 m, no validation has been made; nevertheless, most of the archaeological sites and wooden shipwrecks in Israel lie below this depth. SDB validation is a challenging task since it depends on the appropriate logistics, weather conditions, signal communications, provisions for the team (e.g., food and water), shifts between divers, and constant supervision to maintain consistency between each measurement. For instance, the two validated lines took slightly over 2 h, and the whole validation campaign required approximately 5 h from arrival to departure.

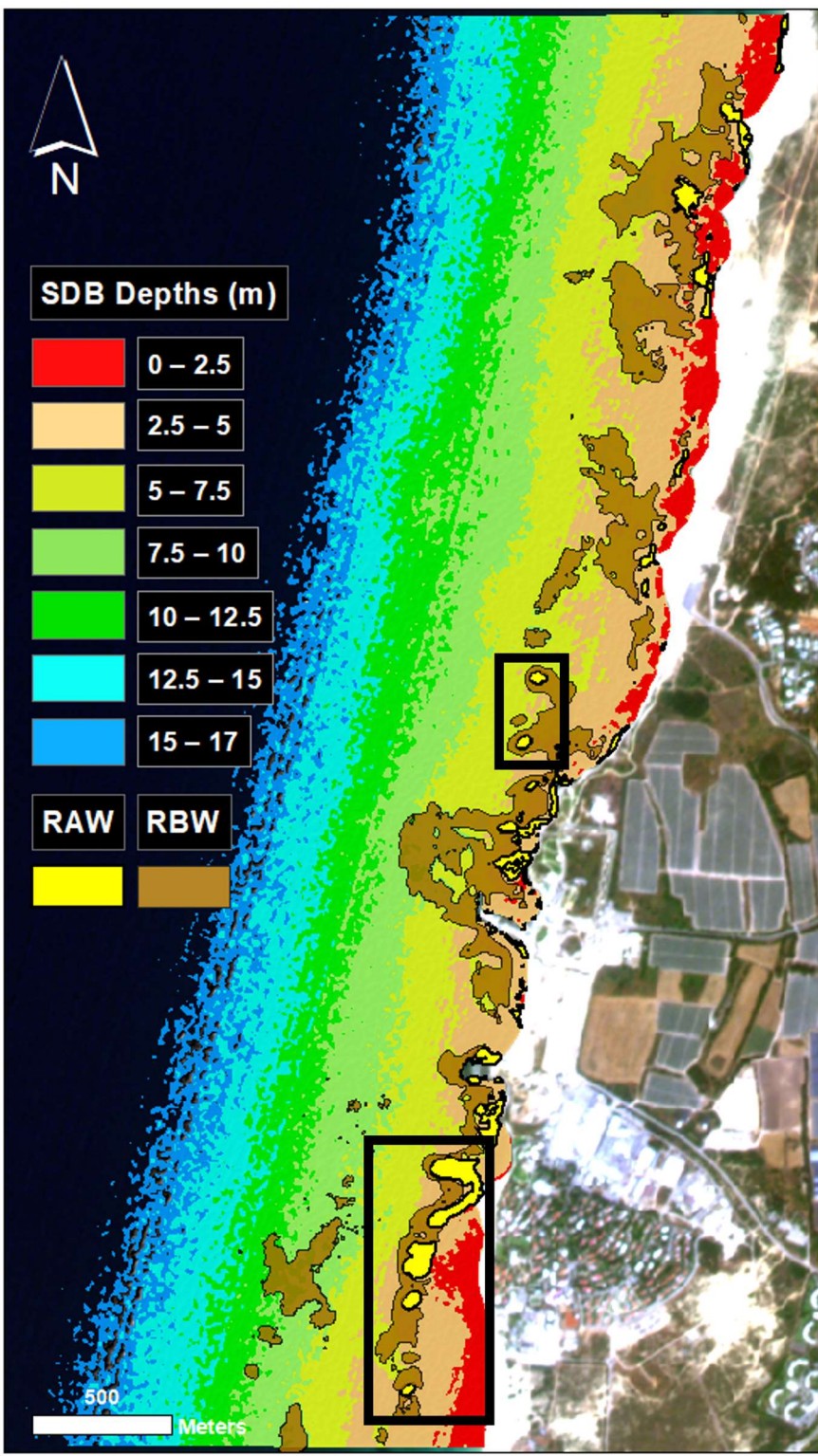

**Figure 5.** Satellite-derived bathymetry map of the Caesarea Maritima area and rock identification map; abrasion platforms are enclosed in the black boxes. Areas identified as RAW and RBW are marked in yellow and brown colours, respectively.

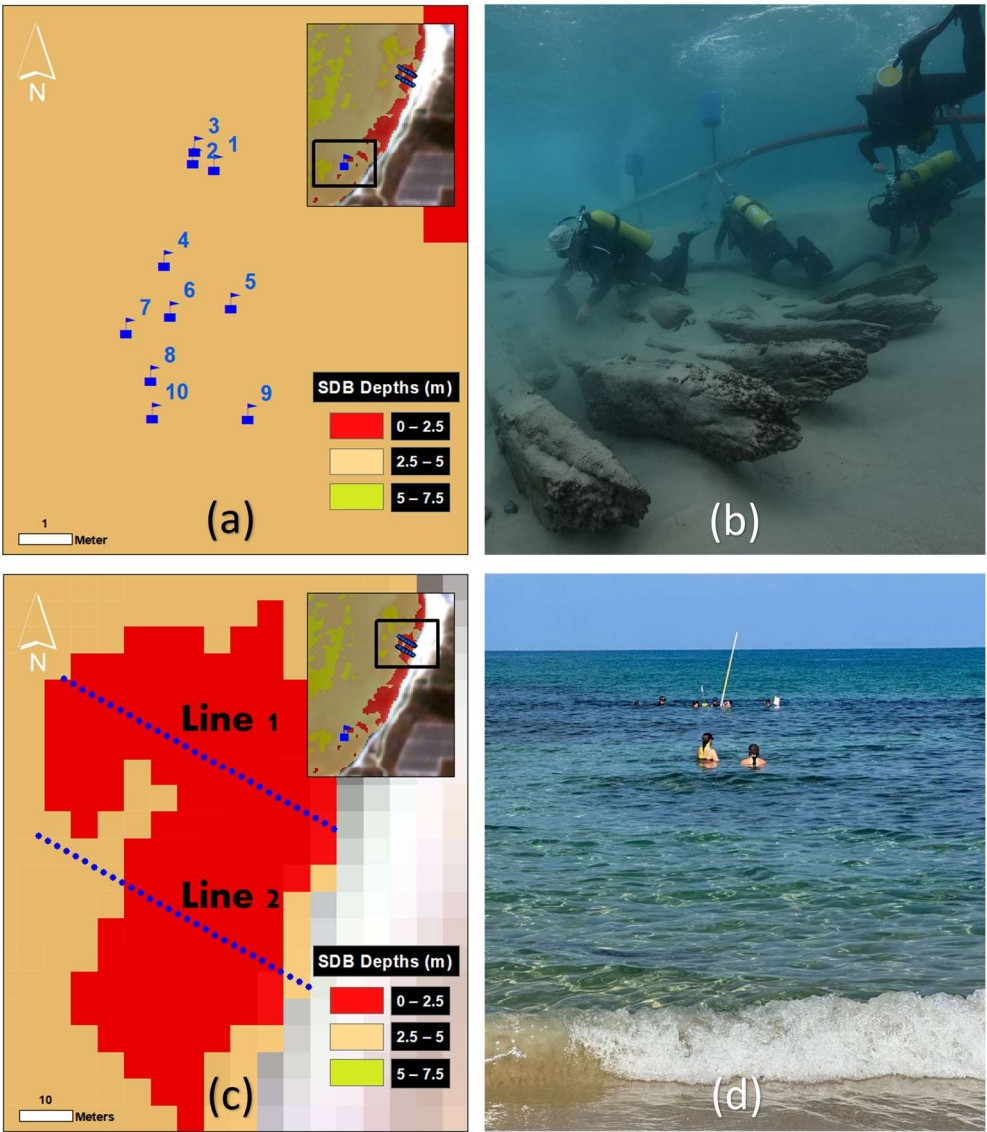

**Figure 6.** Sampling sites of the in situ depth measurements. (**a**) Distribution of depth measurements (numbered blue points, as shown in Table 3) recorded using the total station taken in the vicinity of the shipwreck's hull in 2017. (**b**) An underwater picture of the shipwreck site taken during the survey (photo by Nicolas Ponzone). (**c**) Distribution of 30 depth measurements (blue points) taken along two lines in 2023. (**d**) A picture taken during the 2023 survey, showing a 5 m levelling staff at the time of measurements (photo by Emmanuel Nantet). In panels a and c, the coloured background marks the SDB values, and the inserts show the location of the sampling area.

**Table 3.** Comparison between in situ measurements and SDB values.

| ID | Shipwreck's Hull (m) | SDB (m) | Diff. (%) | ID | Line 1 (m) | SDB (m) | Diff. (%) | Line 2 (m) | SDB (m) | Diff. (%) |
|---|---|---|---|---|---|---|---|---|---|---|
| | | | | 1 | 0.385 | 2.017 | 423.896 | 0.42 | 2.52 | 500.000 |
| 1 | 2.544 | 2.604 | 2.358 | 2 | 0.6 | 2.017 | 236.167 | 0.65 | 2.52 | 287.692 |
| | | | | 3 | 0.83 | 2.017 | 143.012 | 0.83 | 2.52 | 203.614 |
| | | | | 4 | 1.06 | 1.754 | 65.472 | 1.03 | 2.323 | 125.534 |
| 2 | 2.577 | 2.604 | 1.048 | 5 | 1.145 | 1.754 | 53.188 | 1.17 | 2.323 | 98.547 |
| | | | | 6 | 1.355 | 1.859 | 37.196 | 1.31 | 2.323 | 77.328 |
| | | | | 7 | 1.489 | 1.928 | 29.483 | 1.44 | 2.195 | 52.431 |
| 3 | 2.798 | 2.604 | 6.934 | 8 | 1.589 | 1.928 | 21.334 | 1.54 | 2.264 | 47.013 |
| | | | | 9 | 1.691 | 1.928 | 14.015 | 1.58 | 2.195 | 38.924 |
| | | | | 10 | 1.7 | 2.188 | 28.706 | 1.64 | 2.107 | 28.476 |
| 4 | 2.665 | 2.579 | 3.227 | 11 | 1.656 | 2.188 | 32.126 | 1.67 | 2.107 | 26.168 |
| | | | | 12 | 1.73 | 2.188 | 26.474 | 1.8 | 2.107 | 17.056 |
| | | | | 13 | 1.79 | 2.155 | 20.391 | 1.72 | 1.981 | 15.174 |
| 5 | 2.523 | 2.579 | 2.220 | 14 | 1.78 | 2.321 | 30.393 | 1.91 | 1.983 | 3.822 |
| | | | | 15 | 1.815 | 2.321 | 27.879 | 2.72 | 1.983 | 27.096 |
| | | | | 16 | 1.635 | 2.248 | 37.492 | 2.185 | 2.169 | 0.732 |
| 6 | 2.852 | 2.579 | 9.572 | 17 | 1.875 | 2.248 | 19.893 | 2.25 | 2.426 | 7.822 |
| | | | | 18 | 1.85 | 2.317 | 25.243 | 2.29 | 2.169 | 5.284 |
| | | | | 19 | 1.95 | 2.286 | 17.231 | 2.26 | 2.576 | 13.982 |
| 7 | 2.809 | 2.633 | 6.266 | 20 | 1.925 | 2.317 | 20.364 | 2.28 | 2.481 | 8.816 |
| | | | | 21 | 1.935 | 2.28 | 17.829 | 2.28 | 2.426 | 6.404 |
| | | | | 22 | 1.65 | 2.26 | 36.970 | 2.32 | 2.576 | 11.034 |
| 8 | 2.548 | 2.663 | 4.513 | 23 | 1.55 | 2.286 | 47.484 | 2.37 | 2.576 | 8.692 |
| | | | | 24 | 1.17 | 2.286 | 95.385 | 2.36 | 2.576 | 9.153 |
| | | | | 25 | 2.17 | 2.284 | 5.253 | 2.02 | 2.77 | 37.129 |
| 9 | 2.178 | 2.579 | 18.411 | 26 | 2.05 | 2.189 | 6.780 | 1.615 | 2.56 | 58.514 |
| | | | | 27 | 2.37 | 2.189 | 7.637 | 2.795 | 2.56 | 8.408 |
| | | | | 28 | 2.01 | 2.285 | 13.682 | 2.645 | 2.77 | 4.726 |
| 10 | 2.770 | 2.633 | 4.946 | 29 | 2.4 | 2.285 | 4.792 | 2.71 | 2.9 | 7.011 |
| | | | | 30 | 2.4 | 2.558 | 6.583 | 1.72 | 2.77 | 61.047 |

**Table 4.** Statistics of the comparison between in situ measurements and SDB values.

| Statistics | Shipwreck's Hull (10 Points) | Line 1 (30 Points) | Line 2 (30 Points) | All Measurements (70 Points) |
|---|---|---|---|---|
| Mean (m) | 0.513 | 0.531 | 0.615 | 0.513 |
| Std Dev. (m) | 0.109 | 0.364 | 0.55 | 0.46 |
| RMSE (m) | 0.187 | 0.643 | 0.825 | 0.688 |

## 5. Discussion

Providing complementary information to sonar-based bathymetric mapping, the approach herein presented has a number of advantages that are critical in terms of their maritime archaeological implications: (1) augmenting the resolution of the bathymetric map both horizontally and vertically; (2) extending the coverage regions proximate to the shoreline in a cost-effective manner; and (3) displaying a better and more comprehensive illustration of the near-shore landscape by adding RAW and RBW. To recap, a first-stage comparison between the ground-truth sonar and satellite data defined the maximum depth SDB is able to reach. For this study, depths below 17 m yielded an RMSE of 0.949 m, which can be useful and utilitarian for bathymetric mapping on a daily or weekly basis.

Conversely, depths exceeding this threshold underwent an increase in the RMSE up to 2.037 m, as shown in Table 2. The quadratic fitting model (Figure 2c and Equation (2)) allowed for the conversion of the band ratio pixel values into depths for the areas where data were absent. When these SDB depth values were plotted against the in situ measurements in shallow regions, they gave an RMSE of 0.688 m (Table 4). Given the documented coastal variation of ±0.4 m along the Israeli coast [75], this outcome can be deemed highly satisfactory.

The rock identification approaches proved to be efficient and profitable, and when integrated with SDB, they facilitated a more detailed and complete understanding of

the area of interest. Extracting information on the RAW was a straightforward method, requiring only bands 7 and 8 (Figure 3a–c) and setting the reclassification threshold at three times the standard deviation (negative), far from the mean after subtracting band 8 from band 7. In contrast, the RBW required a more toilsome approach, involving a PCA of five pre-selected bands, followed by MLC and an image reclassification (Figure 3d–f). Bands 2, 3, 4, 7, and 8 worked effectively in the PCA process due to a prior and quick analysis of the raw imagery (Figure 3a). This process is not limited to these bands only, and using other satellites will require a similar analysis; however, it is suggested that two of the selected bands provide information about RAW as bands 7 and 8 did, while the other three should display information on RBW. This signifies that, at the very least, some of the RBW should be observable without any processing. Five different classes were needed for the MLC; less than that can lead to an RBW misclassification. Lastly, spectral similarities among the RBW and deeper water were erroneously classified into the same group, which represents a limitation of the method, caused by deeper waters.

Additionally, Figure 5 also shows that most of the study area is characterised by the presence of RBW, mainly at depths ranging from 2.5 to 5 m. This observation holds significant importance in the context of ship drafts, which are intrinsically related to vessel tonnages, which, in turn, offer insights into ships' dimensions by mutually studying the size of their wooden pieces [76]. Ergo, the presence of RAW as abrasion platforms, RBW, and bathymetry data can be used (1) as a convenient tool for studying the sailing trajectories of ancient ships as well as trade networks; (2) for determining suitable areas for underwater surveying and excavation campaigns; and (3) for studying natural anchorages and geomorphological features, helping researchers understand how vessels anchored in various locations, among other applications encompassing similar purposes.

This methodology is of great interest, particularly in the case of this harbour system extending over a large area comprising numerous mooring spots and anchorages. It offers enhanced insights into the Caesarea's harbour system by providing comprehensive bathymetric data not only offshore but also in the shallow areas close to the beach, especially in the abrasion platforms to the north and south of the Herodian port, where the depths are often between 2.5 and 5 m and are cluttered with architectural debris and reefs, which makes them more challenging to access using conventional instruments.

In such dynamic environments where frequent sand shifting leads to notable bathymetric alterations, this cost-effective and easy tool aids archaeologists in efficiently planning surveys and excavations, often at short notice and involving extensive logistical coordination with numerous divers. For instance, during the 2017 excavation season at the Caesarea shipwreck site, the removal of substantial quantities of sand constituted a considerable challenge after a storm event that deposited an abundance of sediments. Conversely, the 2018 season was much easier, as the site was exposed, greatly facilitating the study. If we had readily available SDB in 2017, conducting the excavations would have been much easier.

This methodology also contributes to understanding past bathymetric variations at a given date. The resolution, however, will be dependent on that of the satellite imagery. A pan-sharpening process could enhance the spatial resolution; nonetheless, this could introduce distortions or fake artifacts. It is important to mention that the 5 m resolution SDB map cannot distinguish the 13 m long Caesarea hull from its surrounding context. However, this resolution is high enough to detail more thoroughly the extension of large harbour structures, such as the 500 m long southern breakwater closing the Herodian harbour, visible in some aerial photos.

Consequently, this methodology contributes to reconstructing the harbour landscape at a broad scale, as it highlights the importance of numerous reefs (RAW and RBW), often partly submerged, that close the northern bay almost entirely, showing that this area, although dangerous and challenging to access, could have provided ships with protected shelter. Therefore, close examination of detailed bathymetric data, along with systematical surveys, reveals that several natural bays, not only the Herodian basin on

which archaeologists have focused, played a significant role in the Caesarea's harbour system. Nevertheless, the current bathymetric map (Figure 5) does not provide any clues about the harbour's depth in antiquity, as sea-level changes, combined with seismic activity, have strongly impacted the port landscape, a topic which is beyond the scope of this study.

## 6. Concluding Remarks

This paper demonstrates the potential of integrating low-resolution sonar data with multispectral satellite imagery to improve both the horizontal and vertical resolution while also expanding the area coverage. By combining these data sources, reliable bathymetric mapping up to 17 m depth was achieved.

Furthermore, the subtraction of band 7 from band 8 in the VENμS satellite imagery proved effective in highlighting the rocks above water (RAW). Likewise, bands 2, 3, 4, 7, and 8 successfully identified the rocks below water (RBW) through Principal Component Analysis (PCA) and Maximum Likelihood Classification (MLC), with the potential for other bands' utilisation. These results helped us to comprehend the geological and submerged cultural landscape of the ancient harbour of Sebastos, with particular reference to the abrasion platforms in the region.

Satellite-derived bathymetry is an important issue in heritage management, especially along coasts with significant sand shifting, like in Israel. Its level of detail will depend on the satellite's spatial resolution. This methodology serves as a preliminary step before deploying more specialised remote sensing equipment such as side-scan sonars and sub-bottom profilers, as it is less expensive (e.g., free data), less time-consuming, and theoretically unlimited in its spatiotemporal coverage. In other words, it offers an additional tool that can be easily combined with different techniques.

Additionally, it offers a replicable framework applicable in diverse regions using multispectral satellite imagery, preferably high-resolution, contributing to our ability to reconstruct submerged cultural landscapes. Challenges may arise from factors such as cloud cover, proper satellite imagery band selection, or environmental conditions like turbidity, wave action, sun glint, and chlorophyll content.

Although the approach used in this research is not new in itself, several unique aspects of this work contribute to its novelty and significance within the field of maritime archaeology. To the best of our knowledge this is the first time this particular approach has been applied and validated using VENμS data, which offers distinct advantages compared to data from other non-commercial satellites, such as Sentinel and Landsat.

Importantly, VENμS data are characterised by a remarkably high spatial resolution of 5 meters, enabling detailed bathymetric mapping in the vicinity of archaeological sites, as demonstrated in this study. This enhanced resolution facilitated the identification and mapping of submerged rocks, a capability not commonly achievable using satellite remote sensing. The successful achievement of this task marks a significant advancement in our ability to accurately characterise underwater features and understand their significance within archaeological contexts.

Moreover, VENμS's high temporal resolution (every two days) allows for tracking rapid bathymetric changes due to sand cover, which may have an important contribution to efficient excavation planning. In addition to these methodological advancements, this paper provides the first SDB mapping of the Caesarea city port, offering a fresh perspective on this historically significant archaeological site.

Further studies should be undertaken by testing other very-high-resolution commercial satellite sensors (e.g., Pleiades Neo or WorldView-3), implementing multispectral UAVs to collect more detailed data, investigating the capabilities of SDB and rock identification in different environmental contexts (e.g., lakes and lagoons), or monitoring coastal erosion effects on archaeological sites.

Rock identification should be perfected and optimised to also discern between different rock types or visibly submerged archaeological remains, as should its automation. Lastly,

exploring suitable bands for rock identification using other satellites will be of high interest as well.

**Author Contributions:** G.D., Y.L. and E.N. conceived the project and wrote the paper. G.D. processed and analysed the remote sensing and sonar data. E.N. led the archaeological expeditions. All authors have read and agreed to the published version of the manuscript.

**Funding:** The excavation of the Caesarea shipwreck was funded by the Research Authority of the University of Haifa, the Honor Frost Foundation, and the European Union (ERC, SHIPs, 101088962). The views and opinions expressed are, however, those of the authors only and do not necessarily reflect those of the European Union or the European Research Council. Neither the European Union nor the granting authority can be held responsible for them. Additionally, this research was made possible thanks to the Sir Maurice and Lady Irene Hatter's student scholarship (2021HatterGD).

**Data Availability Statement:** All the data used in this research are available upon request. The satellite data in this research are available for scientific research.

**Acknowledgments:** We would like to express our gratitude to the master's and PhD students from the Department of Maritime Civilizations at the University of Haifa who contributed to this research by collecting the underwater data points. Special thanks to Michael Lazar who provided the sonar data, along with detailed technical specifications, which greatly contributed to this study.

**Conflicts of Interest:** The authors declare no conflicts of interest.

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
