# Peer review of "Satellite-Derived Bathymetry in Support of Maritime Archaeological Research—VENμS Imagery of Caesarea Maritima, Israel, as a Case Study"

_remotesensing, doi:10.3390/rs16071218_

Round 1

Reviewer 1 Report

Comments and Suggestions for Authors

Dear Authors,

The paper is interesting and well structured. However, in my opinion, there are some considerations to be made before it can be ready for publication:

(i) there is a lack of reference on the topic of estimated bathymetry and archaeology which, however, is not a "brand new" practice (e.g., https://www.degruyter.com/document/doi/10.1515/opar-2016-0018/html?lang=en);

(ii) general references to the state of the art should also be increased (e.g., https://www.mdpi.com/2072-4292/14/3/772);

(iii) the conclusions seem to be written too "quickly" (surely not, but they give that impression). Therefore, I would ask you to expand the concluding paragraph, better arguing (i) your work and what implications it may have, (ii) criticalities and strengths of the method used; (iii) future developments based also on new satellites reaching much higher resolutions.

Author Response

February 20, 2024

Dear Editor,

We are delighted to submit a revised version of our manuscript “Satellite derived bathymetry in support of maritime archaeological research - VENμS imagery of Caesarea Maritima, Israel as a case study”, for consideration in the Remote Sensing’s special issue on Remote Sensing Methods and Approaches for Underwater Cultural Heritage Research and Management.

We have thoroughly addressed the constructive comments of the reviewers, which have substantially contributed to the quality of the paper. Importantly, we have recalculated the values in tables 2, 3 and 4, elaborated the concussions section, and added a more thorough description of the methodology and its accuracy and pitfalls. In addition, we realized that the term “northern anchorages” was not accurately used, and it was replaced by the term  “abrasion platforms” throughout the manuscript.

Below please find a point-by-point response to the reviewers' comments.

We hope you find our paper suitable for publication.

On behalf of the coauthors

Gerardo Diaz

Point-by-point response to reviewers

Original comments are included in italics for the sake of clearness.

Reviewer #1

The paper is interesting and well structured. However, in my opinion, there are some considerations to be made before it can be ready for publication

We are pleased to know that the reviewer found our paper interesting and well structured. We are thankful for the useful comments that have contributed to the quality of the paper.

There is a lack of reference on the topic of estimated bathymetry and archaeology which, however, is not a "brand new" practice (e.g., https://www.degruyter.com/document/doi/10.1515/opar-2016-0018/html?lang=en);

We thank the reviewer for bringing this point to our attention. Following this comment, we have added the following paragraphs and associated references:

Lines 61-66: “Maritime archaeologists require high-resolution (in time and space) bathymetric mapping and unbounded spatial coverage for planning underwater prospections, documenting and recording submerged features, performing preservation maneuvers, and conducting excavation campaigns. A few papers can be found in this realm [13,45]; similarly, the European project ITACA uses satellite data for deriving relative bathymetry for the heritage management of coastal archaeological sites [46].”

New references:

  1. Guzinski, R.; Spondylis, E.; Michalis, M.; Tusa, S.; Brancato, G.; Minno, L.; Hansen, L.B. (2016). Exploring the utility of bathymetry maps derived with multispectral satellite observations in the field of underwater archaeology. Open Archaeology. 2016, 2, 243-246.
  2. Final Report Summary - ITACA (Innovation Technologies and Applications for Coastal Archaeological sites). Available online: Innovation Technologies and Applications for Coastal Archaeological sites | ITACA | Project | News & Multimedia | FP7 | CORDIS | European Commission (europa.eu) (Accessed on 09.02.2024).

General references to the state of the art should also be increased (e.g., https://www.mdpi.com/2072-4292/14/3/772).

Following these comments we have added the following paragraph and associated references:

Lines 49-53:  “In recent years, satellite-derived bathymetry (SDB) has been suggested as a reliable, cost-effective alternative [19-21] by means of empirical methods [16,22-25], look-up tables [26-28] or semi-analytical models [29-31]. Lately, stereo approaches [32,33] machine learning [34-36] and deep learning techniques [37] have been implemented as well, allowing for the overcoming of key limitations of conventional approaches.”

New references:

  1. Lyzenga, D.R. Passive remote sensing techniques for mapping water depth and bottom features. Appl. Opt. 1978, 17, 379-383.
  2. Traganos, D.; Reinartz, P. Mapping Mediterranean seagrasses with Sentinel-2 imagery. Mar. Pollut. Bull. 2017, 10, 1-12.
  3. Dierssen, H.M.; Zimmerman, R.C.; Leathers, R.A.; Downes, T.V.; Davis, C.O. Ocean color remote sensing of seagrass and bathymetry in the Bahamas Banks by high-resolution airborne imagery. Limnol. Oceanogr. 2003, 48, 444–455.
  4. Evagorou, E.; Argyriou, A.; Papadopoulos, N.; Mettas, C.; Alexandrakis, G.; Hadjimitsis, D. Evaluation of Satellite-Derived Bathymetry from High and Medium-Resolution Sensors Using Empirical Methods. Remote Sens. 2022, 14, 772.
  5. Hedley, J.D.; Roelfsema, C.; Phinn, S.R. Efficient radiative transfer model inversion for remote sensing applications. Remote Sens. Environ. 2009, 113, 2527–2532.
  6. Mobley, C.D.; Sundman, L.K.; Davis, C.D.; Bowles, J.H.; Downes, T.V.; Leathers, R.A.; Montes, M.J.; Bisset, W.P.; Kohler, D.D.R.; Reid, R.P.; Louchard, E.M.; Gleason, A. Interpretation of hyperspectral remote-sensing imagery by spectrum matching and look-up tables. Appl. Opt. 2005, 44, 3576-3591.
  7. Lee, Z.; Carder, K.L.; Mobley, C.D.; Steward, R.G.; Patch, J.S. Hyperspectral remote sensing for shallow waters: 2. Deriving bottom depths and water properties by optimization. Appl. Opt. 1999, 38, 3831-3843.
  8. Klonowski, W.M.; Fearns, P.R..; Lynch, M.J. Retrieving key benthic cover types and bathymetry from hyperspectral imagery. J. Appl. Remote Sens. 2007, 1, 1-19.
  9. Wettle, M.; Brando, V.E. Sambuca: Semi-analytical model for bathymetry, unmixing and concentration assessment. Tech. Rep. CSIRO Land and Water Sci. Rep. 22/06: Melbourne, Australia, 2006, pp.1-20.
  10. Dekker, A.G.; Phinn, S.R.; Anstee, J.; Bissett, P.; Brando, V.E.; Casey, B.; Fearns, P.; Hedley, J.; Klonowski, W.; Lee, Z.P.; Lynch, M.; Lyons, M.; Mobley, C.; Roelfsema, C. Intercomparison of shallow water bathymetry, hydro-optics, and benthos mapping techniques in Australian and Caribbean coastal environments. Limnol. Oceanogr.-Meth. 2011, 9, 396-425.
  11. Collings, S.; Botha, E.J.; Anstee, J.; Campbell, N. Depth from Satellite Images: Depth Retrieval Using a Stereo and Radiative Transfer-Based Hybrid Method. Remote Sens. 2018, 10, 1247.
  12. Cao, B.; Fang, Y.; Jiang, Z.; Gao, L.; Hu, H. Shallow water bathymetry from WorldView-2 stereo imagery using two-media photogrammetry. Eur. J. Remote Sens. 2019, 52, 506–521.
  13. Sagawa, T.; Yamashita, Y.; Okumura, T.; Yamanokuchi, T. Satellite Derived Bathymetry Using Machine Learning and Multi-Temporal Satellite Images. Remote Sens. 2019, 11, 1155.
  14. Dickens, K.; Armstrong, A. Application of Machine Learning in Satellite Derived Bathymetry and Coastline Detection. SMU Data Sci. Rev. 2019, 2, 1-22.
  15. Tonion, F.; Pirotti, F.; Faina, G.; Paltrinieri, D. A Machine Learning Approach to Multispectral Satellite Derived Bathymetry. ISPRS Ann. Photogramm. Remote Sens. Spat. Inf. Sci. 2020, 3, 565–570.

The conclusions seem to be written too "quickly" (surely not, but they give that impression). Therefore, I would ask you to expand the concluding paragraph, better arguing (i) your work and what implications it may have, (ii) criticalities and strengths of the method used; (iii) future developments based also on new satellites reaching much higher resolutions.

Following this important comment, the conclusions were rewritten as follows:

Lines 591-625: “This paper demonstrates the potential of integrating low-resolution sonar data with multispectral satellite imagery to improve both horizontal and vertical resolution, while also expanding area coverage. By combining these data sources, reliable bathymetric mapping up to 17 meters depth was achieved. Furthermore, the subtraction of Band 7 from Band 8 of VENμS satellite imagery proved effective in highlighting rocks above water (RAW). Likewise, bands 2, 3, 4, 7, and 8 successfully identified rocks below water (RBW) through Principal Component Analysis (PCA) and Maximum Likelihood Classification (MLC), with potential for other bands' utilization.

These results helped to comprehend the geological and archaeological landscape of the ancient harbour of Sebastos, along with the abrasion platforms and the Southern anchorages located in the region. Moreover, the identification of rocks above and below the water is of great value for other disciplines as well (e.g., geological or engineering applications).

Bathymetry is an important issue in heritage management, especially along coasts with significant sand shifting, like in Israel. Frequent storms uncover large areas of the bottom, offering numerous spots for archaeological surveys. This methodology helps planning archaeological operations, such as surveys and excavations, and serves as a preliminary step before deploying more specialized remote sensing equipment such as side-scan sonars and sub-bottom profilers, as it is less-expensive (e.g. free data), less time-consuming, labour non-intensive (e.g. semi-automatic proecedure), and theoretically unlimited in spatiotemporal coverage. In other words, it offers an additional tool that can be easily combined with different techniques.

Additionally, it offers a replicable framework applicable in diverse regions using multispectral satellite imagery, preferably high-resolution, contributing to our ability to perform maritime archaeological research. Challenges may arise from factors such as cloud cover, band selection in satellite imagery, or environmental conditions like turbidity, wave action, sun-glint, and chlorophyll content.

Further studies should be done by testing other high-resolution satellite sensors (e.g., Pleiades Neo or WorldView-3), implementing multispectral UAVs to collect more detailed data, and investigating the capabilities of SDB and rock identification in different environmental contexts (e.g., lakes and lagoons), or monitoring coastal erosion effects on archaeological sites. Rock identification should be perfected and optimized to also discern between different rock types or visibly submerged archaeological remains, as should its automation. Lastly, exploring suitable bands for rock identification using other satellites will be of high interest as well.”

Reviewer 2 Report

Comments and Suggestions for Authors

This work clearly presents the effectiveness of a methodological approach (for the acquisition of near-shore Bathymetry from Satellite imagery), already established in other research fields, in maritime archaeology. Moreover the identification of Rocks Above and Below the Water is of great value for other disciplines as well (e.g. geological or engineering applications).

The main question addressed by this work is whether SDB and rock identification can be applied in maritime archaeology.

Geoinformation methodologies and techniques are becoming a new trend in the field of archaeology. Particularly in the maritime domain, where research prerequisites the previous recognition of the geological and archaeological landscape. This work is rather relevant by providing an approach that can readily be replicated in shallow-water sites and with various multispectral satellite imagery. Moreover, the presented approach covers a gap concerning the implementation of SDB in maritime archaeology, a technique commonly used in other research fields.

Compared to existing remote sensing methodologies for bathymetry (from the water-surface or airborne), this work proposes an alternative that is less-expensive (e.g. free data), less time-consuming, labour non-intensive (e.g. semi-automatic procedure), and theoretically unlimited in spatiotemporal resolution and coverage.

The performance of this methodology was sufficiently evaluated by ground-truthing [lines 407-409], in situ measurements [lines 413-414] and the reliability of the rock identification approach [lines 503-504].

The results (Satellite-Derived Bathymetry and Rock identification) were appropriately validated against data from on-site surveys (ground-truthing during archaeological expedition and drone-based imaging, respectively).

The proposed method is affected by the environmental conditions (sea state, sun-glint, water turbidity, clouds) over the area of interest. The main limitations concern either the depth at which the method is applied (due the nature of light attenuation along the water column), or regarding the expected spatial resolution (dependent on the satellite characteristics).

Beyond the presented work it would be of interest further research on the performance of the SDB and rock identification methodology over a given period, during which environmental conditions may vary (e.g. for monitoring the possible costal erosion effects on archaeological sites).

The concluding remarks [lines 585-588] are consistent with the evidence and arguments presented and they address the main question.

This work presents an extensive and up to date list of references that are appropriately support the research.

Tables and figures are clearly presenting data and the results. 

Author Response

February 20, 2024

Dear Editor,

We are delighted to submit a revised version of our manuscript “Satellite derived bathymetry in support of maritime archaeological research - VENμS imagery of Caesarea Maritima, Israel as a case study”, for consideration in the Remote Sensing’s special issue on Remote Sensing Methods and Approaches for Underwater Cultural Heritage Research and Management.

We have thoroughly addressed the constructive comments of the reviewers, which have substantially contributed to the quality of the paper. Importantly, we have recalculated the values in tables 2, 3 and 4, elaborated the concussions section, and added a more thorough description of the methodology and its accuracy and pitfalls. In addition, we realized that the term “northern anchorages” was not accurately used, and it was replaced by the term  “abrasion platforms” throughout the manuscript.

Below please find a point-by-point response to the reviewers' comments.

We hope you find our paper suitable for publication.

On behalf of the coauthors

Gerardo Diaz

Point-by-point response to reviewers

Reviewer #2

This work clearly presents the effectiveness of a methodological approach (for the acquisition of near-shore Bathymetry from Satellite imagery), already established in other research fields, in maritime archaeology. Moreover the identification of Rocks Above and Below the Water is of great value for other disciplines as well (e.g. geological or engineering applications). The main question addressed by this work is whether SDB and rock identification can be applied in maritime archaeology.

Geoinformation methodologies and techniques are becoming a new trend in the field of archaeology. Particularly in the maritime domain, where research prerequisites the previous recognition of the geological and archaeological landscape. This work is rather relevant by providing an approach that can readily be replicated in shallow-water sites and with various multispectral satellite imagery. Moreover, the presented approach covers a gap concerning the implementation of SDB in maritime archaeology, a technique commonly used in other research fields.

Compared to existing remote sensing methodologies for bathymetry (from the water-surface or airborne), this work proposes an alternative that is less-expensive (e.g. free data), less time-consuming, labour non-intensive (e.g. semi-automatic procedure), and theoretically unlimited in spatiotemporal resolution and coverage.

The performance of this methodology was sufficiently evaluated by ground-truthing [lines 407-409], in situ measurements [lines 413-414] and the reliability of the rock identification approach [lines 503-504]. The results (Satellite-Derived Bathymetry and Rock identification) were appropriately validated against data from on-site surveys (ground-truthing during archaeological expedition and drone-based imaging, respectively). The proposed method is affected by the environmental conditions (sea state, sun-glint, water turbidity, clouds) over the area of interest. The main limitations concern either the depth at which the method is applied (due the nature of light attenuation along the water column), or regarding the expected spatial resolution (dependent on the satellite characteristics). Beyond the presented work it would be of interest further research on the performance of the SDB and rock identification methodology over a given period, during which environmental conditions may vary (e.g. for monitoring the possible costal erosion effects on archaeological sites).

The concluding remarks [lines 585-588] are consistent with the evidence and arguments presented and they address the main question. This work presents an extensive and up to date list of references that are appropriately support the research.

 We are glad to know that the reviewer found the paper to be robust and suitable for publication, and are thankful for the kind words of appreciation. Following the reviewer comments the following sentences were added to the conclusions:

Lines 601-603: “Moreover, the identification of rocks above and below the water is of great value for other disciplines as well (e.g. geological or engineering applications).”

Lines 609-611: “[...] as it is less-expensive (e.g. free data), less time-consuming, labour non-intensive (e.g. semi-automatic procedure), and theoretically unlimited in spatiotemporal coverage.”

Reviewer 3 Report

Comments and Suggestions for Authors

The manuscript presents a high resolution image of the Vegetation and Environment monitoring New Micro-Satellite (VENμS), and attempts to combine sonar data of a coarse resolution to achieve bathymetry retrieval and rock identification. Satellite bathymetry and archaeological research are mentioned in the title, however, the content of archaeology involved in the whole manuscript is quite limited. Questions as how the water depth affects maritime archaeological research and what scale or resolution of depth data is needed should be expressed somewhere. And here are the following issues that need to be addressed.

Line 167: Is it ≤40 or >40.

Section 3.2: The sonar data were collected in 2001 and 2006, which is far from the image acquisition time of 2020. The compatibility of the two should be explained.

Section 3.3: What instrument or equipment was used to measure the ground-truth data of water depth, and how much can the sounding accuracy be achieved.

Equation 1: What is n present and what is its value.

Lines 243~244: Which resampling technique is chosen. What I doubt is that although the image are down sampled to the same spatial resolution, they should not be exactly aligned, how to lessen the offset between the pixels of the two images.

Equation 2 & Line 249: Whether x is uppercase or lowercase.

Lines 254~256: Is the final SDB map upsampling by interpolation? Wouldn't it be more accurate to interpolate the sonar results directly?

Figure 2: 1) No unit for depth. 2) The formula in Figure c is different from Equation 2, rounding or outright truncation should be used to preserve decimals rather than both. Currently, the standards for determining values of the three parameters are different. 3) The sonar data must have been processed, otherwise why are the depths all round numbers?

Table 2: 1) Difference in percentage should be the ratio of the difference to the truth value, not to the SDB or mean value. 2) The RMSE is sensitive to outliers. But you compare the mean value of the SDB with the sonar data, the resulting RMSE cannot represent the reliability of the entire result. The above problems also exist in Table 3

Section 3.5: The types are not appropriate. Since the sea/land mask has been made, there should not be sandy shore which belong to one type of land features.

Figure 4: Is this taxonomy correct? As we can see, the RAW isn’t a pure rock, and there are a lot of undivided pixels at the edge between RBW(brown) and RAW(yellow).

Table 3: 1) There are a lot of misaligned characters in the table, and the fonts are not uniform. 2) Although the error appears to be small, the deepest measured depth does not exceed 3 meters, which can not mean that the SDB is accurate.

Line 608: Wrong punctuation at the end.

Author Response

February 20, 2024

Dear Editor,

We are delighted to submit a revised version of our manuscript “Satellite derived bathymetry in support of maritime archaeological research - VENμS imagery of Caesarea Maritima, Israel as a case study”, for consideration in the Remote Sensing’s special issue on Remote Sensing Methods and Approaches for Underwater Cultural Heritage Research and Management.

We have thoroughly addressed the constructive comments of the reviewers, which have substantially contributed to the quality of the paper. Importantly, we have recalculated the values in tables 2, 3 and 4, elaborated the concussions section, and added a more thorough description of the methodology and its accuracy and pitfalls. In addition, we realized that the term “northern anchorages” was not accurately used, and it was replaced by the term  “abrasion platforms” throughout the manuscript.

Below please find a point-by-point response to the reviewers' comments.

We hope you find our paper suitable for publication.

On behalf of the coauthors

Gerardo Diaz

Point-by-point response to reviewers

Reviewer #3

The manuscript presents a high resolution image of the Vegetation and Environment monitoring New Micro-Satellite (VENμS), and attempts to combine sonar data of a coarse resolution to achieve bathymetry retrieval and rock identification. Satellite bathymetry and archaeological research are mentioned in the title, however, the content of archaeology involved in the whole manuscript is quite limited. Questions as how the water depth affects maritime archaeological research and what scale or resolution of depth data is needed should be expressed somewhere. And here are the following issues that need to be addressed.

We thank the reviewer for bringing this important point to our attention. Following this comment the following paragraphs were added to the manuscript:

Lines 604-612: “Bathymetry is an important issue in heritage management, especially along coasts with significant sand shifting, like in Israel. Frequent storms uncover large areas of the bottom, offering numerous spots for archaeological surveys. This methodology helps planning archaeological operations, such as surveys and excavations, and serves as a preliminary step before deploying more specialized remote sensing equipment such as Side-scan sonars (SSS) and Sub-bottom profilers (SBP), as it is less-expensive (e.g. free data), less time-consuming, labour non-intensive (e.g. semi-automatic proecedure), and theoretically unlimited in spatiotemporal coverage. In other words, it offers an additional tool that can be easily combined with different techniques.”

Line 167: Is it ≤40 or ï¼ž40.

It should be  ≤40, and was corrected accordingly.

Section 3.2: The sonar data were collected in 2001 and 2006, which is far from the image acquisition time of 2020. The compatibility of the two should be explained.

Following this important comment, a discussion on the compatibility of the datasets has been added (lines 210-217): “It is important to note that while the sonar bathymetry data is used as a benchmark for regional bathymetry, it does not take into account temporal changes associated with different processes such as sea level rise, sediment transport, and coastal erosion. While the time difference between the sonar and satellite data collection is of the order of 20 years, given the fact that the sonar is collected at water depths larger than 5 m, it is likely to assume that during that period bathymetry would not have exhibited large-scale changes that are manifested in the relatively coarse vertical and spatial resolution sonar data.”

Section 3.3: What instrument or equipment was used to measure the ground-truth data of water depth, and how much can the sounding accuracy be achieved.

The instruments that were used to measure the ground-truth data and their accuracy are now described in details in lines 194-200:

Sonar bathymetric data was obtained using a Kongsberg EM 1002 multibeam echosounder, placed onboard the R/V Eziona, and the ELAC SeaBeam 3050 N multibeam system from Wärtsilä, deployed onboard the R/V Mediterranean Explorer [61]. The sounding depth total accuracy for the EM 1002 multibeam is approximately 10   cm in shallow waters; the expected total system accuracy is 0.2% of the depth (from vertical up to 45 degrees), 0.3% of the depth (up to 60% degrees) and 0.5%  of the depth (between 60 and 70 degrees) [62]. Seabeam 3050 N accuracy is down to 2 cm [63].

New references:

  1. EM 1002 – Multibeam echo sounder. Available online: http://linux.geodatapub.com/shipwebpages/survey%20gear/Multibeam/EM1002%20-%20Powell/M%201002%20Product%20Description.pdf (Accessed on 14.02.2024).
  2. Seabeam 3050 N at a Glance. Available online: https://www.yumpu.com/en/document/read/27114766/seabeam-3050-n-elac-nautik/4 (Accessed on 15.02.2024).

Equation 1: What is n present and what is its value.

The value of n is 1000, and it is included in the equation to ensure a positive outcome. The following text and references were added to clarify that.

Lines 255-256: “Where λ_i and λ_j correspond to the blue and green wavelengths, respectively; n is generally set to 1000 to ensure a positive outcome [18,19,65].”

Previous references:

  1. Traganos, D.; Poursanidis, D.; Aggarwal, B.; Chrysoulakis, N.; Reinartz, P. Estimating satellite-derived bathymetry (SDB) with the google earth engine and sentinel-2. Remote Sens. 2018, 10, 859.
  2. Jagalingam, P.; Akshaya, B.J.; Hegde, A.V. Bathymetry Mapping Using Landsat 8 Satellite Imagery. Procedia Eng. 2015, 116, 560-566.

New references:

  1. Lyzenga, D.R. Passive remote sensing techniques for mapping water depth and bottom features. Appl. Opt. 1978, 17, 379-383.

Lines 243~244: Which resampling technique is chosen. What I doubt is that although the image are down sampled to the same spatial resolution, they should not be exactly aligned, how to lessen the offset between the pixels of the two images.

Following this comment we elaborate the description of the resampling method and add an estimation of the error. Specifically, the following text was added:

Lines 264 - 268: “Then, all pixel values were resampled from 5 to 50 m pixel size to match the sonar’s spatial resolution using the nearest resampling technique, since it is the fastest resampling technique and it does not change the cells values. The maximum spatial error is one-half the cell size [66], which does not pose a significant difference for the following processing stages.”

New reference:

  1. Resample (Data Management). Available online: https://pro.arcgis.com/en/pro-app/latest/tool-reference/data-management/resample.htm (accessed on 19.02.2024).

Equation 2 & Line 249: Whether x is uppercase or lowercase.

x should be uppercase ‘X’. It has been corrected accordingly. Now is in line 274.

Lines 254~256: Is the final SDB map upsampling by interpolation? Wouldn't it be more accurate to interpolate the sonar results directly?

The interpolation used for constructing the final SDB map is done for upscaling the 5 m resolution satellite data, which is much more spatially detailed than the 50 m resolution sonar data. Creating a bathymetry map by only interpolation of the 50 m resolution sonar data would have led to a “pseudo high-resolution” map, in which interpolation is filling much larger areas.

Figure 2: 1) No unit for depth. 2) The formula in Figure c is different from Equation 2, rounding or outright truncation should be used to preserve decimals rather than both. Currently, the standards for determining values of the three parameters are different. 3) The sonar data must have been processed, otherwise why are the depths all round numbers?

Meters have been added in Figs. 2, 5 and 6, since interpolation was used to produce the final SDB map, the Equation of Fig. 2.c is kept, and Equation 2 has been modified accordingly.  The sonar datafile (already processed when received) has a vertical resolution of 1 m. Here lies the importance of the method not only increasing its spatial resolution in X-Y from 50 m to 5 m but also its vertical one from 1 m to cm variations. This is explained in the paper (lines 201-203).

Table 2: 1) Difference in percentage should be the ratio of the difference to the truth value, not to the SDB or mean value. 2) The RMSE is sensitive to outliers. But you compare the mean value of the SDB with the sonar data, the resulting RMSE cannot represent the reliability of the entire result. The above problems also exist in Table 3.

Values have been corrected accordingly. In Table 4, Std Dev for shipwreck hull, line 1 and 2 changed because they were mistakenly calculated using the Sample Standard Deviation; the complete set of measurements (70) did not change since Population Standard Deviation was used for it. Line 2 “mean” changed too because I inadvertently exchanged the Mean and Std Dev values. All calculations were done again to ensure that no more typos are present.

Section 3.5: The types are not appropriate. Since the sea/land mask has been made, there should not be sandy shore which belong to one type of land features.

This point has been corrected - sandy shores have been replaced for swash zones according to the beach type of Caesarea.

Figure 4: Is this taxonomy correct? As we can see, the RAW isn’t a pure rock, and there are a lot of undivided pixels at the edge between RBW(brown) and RAW(yellow).

This is due to the pixel size of the images. VENμS imagery from which RAW and RBW have been obtained are 5 m resolution. The drone orthomosaic pixel size lies within the range of centimeters. Evidently, had VENμS had the same spatial resolution as that of the orthomosaics, these gaps would not exist. This is mentioned in lines 365-367.

Table 3: 1) There are a lot of misaligned characters in the table, and the fonts are not uniform. 2) Although the error appears to be small, the deepest measured depth does not exceed 3 meters, which can not mean that the SDB is accurate.

1) The table was realigned and the fonts were corrected. 2) We thank the reviewer for drawing our attention to this important point. Indeed, the data  presented in table 3 are meant only for validating the bathymetry measurements at water depths associated with our archaeological surveys, which are limited to water depths smaller than 3 meters and are not covered by the sonar dataset. We agree with the reviewer that these measurements can be used for validating SDB at larger depths, and there is a range of depths, between 3 and 6 m, which are not covered in our validation. To clarify this point, the text has been modified as follows:

Line 416-424:  “We note that while the SDB values have been well validated for water depths below 3 m (Fig. 6) and between 6 and 17 m (Fig. 2.d), for the depth range of 3-6 m, no validation has been made, nevertheless, most of the archaeological sites and wooden shipwrecks in Israel lie below this depth. SDB validation is a challenging task since it depends on appropriate logistics, weather conditions, signal communications, provisions for the team (e.g., food and water), shifts between divers, and constant supervision to maintain consistency between each measurement. For instance, the two validated lines took slightly over 2 hours, and the whole validation campaign required approximately 5 hours from arrival to departure.”

Line 608: Wrong punctuation at the end.

This has been corrected

Round 2

Reviewer 3 Report

Comments and Suggestions for Authors

I still have some questions about Table 2 and the calculation process.

Are the sonar data used to derive the Equation 1 the same as the data in Table 2? If so, it does not effectively verify the accuracy of SDB over 6m. Because the truth data for modeling and validation should not overlap.

In addition, the RMSE of 6-17m and 6-24m in Table 2 are represented in parentheses and brackets, respectively. It is a typo or you really want to make a difference.

The RMSE is sensitive to outliers. But after averaging, the outliers no longer stand out. You compare the “mean” value of the SDB with sonar data, while the outliers with significant differences have already been eliminated by the average processing, how can the resulting RMSE represent the reliability of the entire result.

Author Response

February 25, 2024

Dear Editor,

We are delighted to submit a revised version of our manuscript “Satellite derived bathymetry in support of maritime archaeological research - VENμS imagery of Caesarea Maritima, Israel as a case study”. 

Below please find a point-by-point response to the important comments of reviewer #3. 

We hope you find our paper suitable for publication.

On behalf of the coauthors

Gerardo Diaz

I still have some questions about Table 2 and the calculation process.

Are the sonar data used to derive the Equation 1 the same as the data in Table 2? If so, it does not effectively verify the accuracy of SDB over 6m. Because the truth data for modeling and validation should not overlap.

The derivation of the Equation 2 and the validation shown in Table 2 were performed in two different regions, which do not overlap. In order to clarify this important point the text was modified as follows:

Lines 274-278. “With X being the band ratio described in equation 1. Subsequently, this equation was validated by comparing the estimated satellite-derived depths and the corresponding sonar data over the port of Hadera, which is found north of the area used for deriving the SDB equation (delineated by a golden frame in Fig. 2b). This comparison yielded  a correlation coefficient (R2) of 0.9355 (Fig. 2d).”

Lines 298-301 (caption of Table 2):  “Comparison between sonar and SDB depths over the area used for validation of SDB equation in the port of Hadera (golden framed box in Fig. 2b). The satellite data is resampled to fit the spatial resolution of the sonar data (of 50 m/pixel). RMSE values are calculated from the entire datasets.”

In addition, the RMSE of 6-17m and 6-24m in Table 2 are represented in parentheses and brackets, respectively. It is a typo or you really want to make a difference.

This is a typo. Brackets are used for both: [6-17] and [6-24].

The RMSE is sensitive to outliers. But after averaging, the outliers no longer stand out. You compare the “mean” value of the SDB with sonar data, while the outliers with significant differences have already been eliminated by the average processing, how can the resulting RMSE represent the reliability of the entire result.

We are thankful to the reviewer for pointing out this important issue. Following this comment, we have recalculated the RMSE over the entire dataset, and modified the table and the text accordingly.